# Complications during Pregnancy after Abdominal Burn Scars: A Review

**Zosha J. van Gelder** [1], **Annabel Snoeks** [2], **Paul P.M. van Zuijlen** [2,3,4,5], **Ralph de Vries** [6] and **Anouk Pijpe** [2,4,7,*]

1 Amsterdam University Medical Center, 1105 AZ Amsterdam, The Netherlands
2 Burn Center, Red Cross Hospital, 1942 LE Beverwijk, The Netherlands
3 Department of Plastic and Reconstructive Surgery, Red Cross Hospital, 1942 LE Beverwijk, The Netherlands
4 Department of Plastic, Reconstructive and Hand Surgery, Amsterdam Movement Sciences, 1081 HZ Amsterdam University Medical Center Amsterdam, The Netherlands
5 Pediatric Surgical Centre, Emma Children's Hospital, Amsterdam University Medical Center, 1105 AZ Amsterdam, The Netherlands
6 Medical Library, Vrije Universiteit, 1081 HV Amsterdam, The Netherlands
7 Association of Dutch Burn Centres, Red Cross Hospital, 1941 AJ Beverwijk, The Netherlands
* Correspondence: apijpe@rkz.nl; Tel.: +31-25-126-5734

**Abstract:** Over the past decades, long-term sequelae of burns have gained increasing attention. Women of childbearing age, who sustained abdominal burns earlier in life, may have unmet information needs on scar-related complications they can expect during pregnancy. We performed a review of the literature to identify abdominal, foetal, and potential other complications during pregnancy in women with abdominal burn scars. PubMed, Embase, and Scopus were searched from inception to 1 July 2020 and updated once on 23 April 2021 (PROSPERO CRD42022187883). Main search terms included pregnancy, scar, burns, and abdominal. Studies on burns obtained during pregnancy have been excluded. Screening, data extraction and bias assessment were conducted by two investigators. We included 22 studies comprising 217 patients. The time between burn injury and first pregnancy varied between 7 and 32 years. Most of the women had normal pregnancies regarding delivery mode and duration of pregnancy. The most reported abdominal burn scar complications were an increased feeling of tightness, itch, pain, and scar breakdown. In some cases, scar release surgery was performed during or prior to pregnancy. Some cases of foetal complications were described. Complications during pregnancy after abdominal burn scars may be limited. More quantitative and qualitative research is needed to assess the maternal and foetal outcomes and complications. The results may be used to inform women and contribute to personalised obstetric management.

**Keywords:** information; burns; pregnancy; scars; complications; abdominal; foetal





## 1. Introduction

Over the years, burn survival has improved in high income countries because there have been numerous improvements in treatments, surgical critical care, a multidisciplinary approach, and surgical interventions [1,2]. The focus of burn care has therefore shifted to survivorship and long-term sequelae of burns, such as scar quality and quality of life [3]. Based on anecdotal input from patients, we learned that women of childbearing age, who sustained severe abdominal burns earlier in life, have unmet information needs on if, when, and which scar-related complications they can expect during pregnancy (Figure A1). Although these women may be a rare subgroup of patients, the impact and long-term consequences can be significant and may even affect their decision to have children. Severe burns of the abdomen may result in scars that may restrict chest and abdominal wall expansion [4]. These scars can lead to multiple complications, such as disfigurement and breathing difficulties.

Pregnancy is accompanied by profound adaptations. This makes pregnant women susceptible to changes in skin and appendages, both physiologic and pathologic, such as

infections, probably due to low cellular immunity [5]. Cutaneous alterations during pregnancy are mainly regulated by hormonal, immunologic, and metabolic factors [6]. In the literature there are cases reporting worsening and recurrence of keloids and hypertrophic scars during pregnancy [7–9]. It is hypothesised that the hormonal changes stimulate scar growth; however, there is little evidence in the literature to support this [10].

The aim of this study was to identify and describe potential abdominal, foetal, and other complications during pregnancy in women with abdominal burn scars.

## 2. Materials and Methods

### 2.1. Protocol

Our protocol was developed and registered in PROSPERO (CRD42022187883), the international prospective register of systematic reviews [11]. We started our study in 2020 and upon updating our PROSPERO record in 2022, we learned that the PROSPERO system and website had been revised and renewed registration was necessary. For this reason, the first submission and registration date do not match the actual study procedure. For reporting, we followed the Preferred Reporting Items for Systematic Reviews and Meta-Analyses (PRISMA) statement [12].

### 2.2. Search Strategy

To identify all relevant publications, we conducted systematic searches in the bibliographic databases PubMed, Embase.com, and Scopus from inception to 1 July 2020 at first and updated once, using the same search strategy, to include studies between 1 July 2020 and 23 April 2021, in collaboration with a medical information specialist (RdV). The following terms were used (including synonyms and closely related words) as index terms or free-text words: "Cicatrix", "Scar", "Pregnancy", "Maternal", "Foetal", "Abdominal", "Truncal", and "Burns". Duplicate articles were excluded. The references of the identified articles were searched for relevant publications. The full search strategies for all databases can be found in Appendix A, Tables A1–A3.

### 2.3. Selection Process and Criteria

The screening process was conducted with the use of the web-based software platform Rayyan (www.rayyan.qcri.org, accessed on 1 July 2020), which has been selected as a preferred tool by the Cochrane Collaboration. Two reviewers (Z.J.v.G. and A.S.) independently screened all potentially relevant titles and abstracts, and full text articles for eligibility. Differences in judgement were resolved through a consensus procedure involving a third independent reviewer (A.P). We primarily wanted to identify and describe potential abdominal scar complications that may arise during pregnancy in women with abdominal burn scars.

In addition, we wanted to learn what possible foetal and postnatal complications can be expected after abdominal or thoracic burn scars. Studies were included if they met the following criteria: (i) pregnant women; (ii) (abdominal) burn scars due to burn injuries in the past; (iii) foetal complications; (iv) other complications due to burns. We excluded studies if they concerned: (i) burns located solely on extremities; (ii) burns obtained during pregnancy; (iii) languages other than English; (iv) animal studies; (v) editorials and letters. Regarding burns obtained during pregnancy, a recent study was review published which included the presentation of a comprehensive guideline [13].

### 2.4. Data Extraction

Two reviewers (Z.J.v.G. and A.S.) independently performed the data extraction of the included studies. Data were extracted using a standardised data extraction Excel sheet. The extraction sheet contained study characteristics (study design, country, study time period, number of patients, outcome measures, data source), patient characteristics (age during burn, cause of burn, age at pregnancy, percentage of abdomen affected, burn characteristics, duration of pregnancy), and outcomes (abdominal complications, foetal complications,

mode of delivery, and breastfeeding). Foetal and postnatal complications were described when multiple studies listed these complications as possible complications in pregnancy.

### 2.5. Risk of Bias Assessment

Risk of bias was assessed using design-concordant tools: the Chambers criteria [14] for case series and case reports and Newcastle–Ottawa Scale (NOS) [15] for the cohort study. Two reviewers (Z.J.v.G. and A.S.) independently evaluated the methodological quality of the full text papers. Differences in judgement of data extraction and risk of bias assessment were resolved through a consensus procedure involving a third independent reviewer (A.P.).

### 2.6. Strategy for Data Synthesis

Since our review was descriptive in nature and we expected a limited number of studies with a wide range of outcomes, we present the extracted data on abdominal burn scar and other complications in a descriptive manner, taking into account the heterogeneity in study design and outcome parameter assessment. For our primary outcome parameter, abdominal burn scar complications, we aimed to use the most widely accepted definition to cover all complications, such as subjective complaints and surgical interventions.

## 3. Results
### 3.1. Literature Search

The flow chart of the search and selection process of studies is presented in Figure A2. The literature search generated a total of 4298 references. After removal of 1964 duplicates, 2334 references remained. A search in Cochrane (clinicaltrials.gov and International Clinical Trials Registry Platform, accessed 23 April 2021) showed no additional relevant articles.

A total of 124 full-text articles were reviewed for eligibility including eight articles that were included through a cross reference check. After full-text screening, 22 articles were included. The reasons for excluding were: wrong setting/study (*n* = 70), foreign language (*n* = 22), no relevant outcomes (*n* = 6), no access to the full text (*n* = 3), and animal study (*n* = 1).

### 3.2. Study and Patient Characteristics

Table 1 shows an overview of the study characteristics of the 22 included articles which covered 217 patients, of whom the majority (*n* = 134/217) originated from one observational, comparative registry-based cohort study from Australia. Other studies were retrospective case series (*n*= 7) and case reports (*n* = 14). The first study was published in 1948. Patients originated from all continents.

**Table 1.** Characteristics of included studies (*n* = 22) by size.

| Study | Study Design | Country | Study Time Period | Patients (*n*) | Pregnancies *n* | Outcome Measures ¥ | Data Source | Risk of Bias |
|---|---|---|---|---|---|---|---|---|
| Duke (2012) | Cohort (retrospective) | Australia | 1983–2008 | 134 | 213 | Abdominal, other | MNS * database | Good |
| Rai (1975) | Case series (retrospective) | UK | 1948–1967 | 21 | 42 | Abdominal, other | Burns Unit | Poor |
| Kitzmiller (1998) | Case series (retrospective) | USA | 1975–1989 | 19 | 31 | Abdominal, other | Burns Unit | Poor |

**Table 1.** *Cont.*

| | | | | | | | | |
|---|---|---|---|---|---|---|---|---|
| Mitsukawa (2015) | Case series (prospective) | Japan | 2000–2015 | 12 | - | Abdominal, other | Department of PS ˆ | Poor |
| McCauley (1990) | Case series (retrospective) | USA | 1967–1985 | 7 | 14 | Abdominal, other | Burns Institute | Poor |
| Daw (1983) | Case series (retrospective) | UK | 1976–1981 | 6 | 11 | Abdominal, other | Department of Gynaecology | Poor |
| Matthews (1982) | Review based on personal communication | UK | Unknown | 2 | 2 | Abdominal | Centre for Burns and PS | Poor |
| Widgerow (1991) | Case report | South Africa | 1990 | 2 | 2 | Abdominal | Department of PS | Poor |
| Arabi (2019) | Case report | Malaysia | 2015 | 1 | 1 | Other | Health Clinic | Poor |
| Aykan (2012) | Case report | Turkey | 2012 | 1 | 1 | Abdominal, other | Department of PS | Poor |
| Cox (2015) | Case report | USA | 2015 | 1 | 1 | Abdominal, other | Department of dermatology | Poor |
| Del Frari (2004) | Case report | Austria | Unknown | 1 | 1 | Natural tissue expansion | Department of PS | Poor |
| Digregorio (1993) | Case report | USA | Unknown | 1 | 1 | Natural expansion | Department of PS | Poor |
| Fioretti (1987) | Case report | Italy | Unknown | 1 | 1 | Abdominal, other | Department of Gynaecology | Poor |
| Haeseker (1981) | Case report | Wales | Unknown | 1 | 1 | Abdominal, foetal | Centre of PS | Poor |
| Kakagia (2012) | Case report | Greece | Unknown | 1 | 1 | Abdominal | Department of PS | Poor |
| Ozog (1963) | Case report | USA | 1962 | 1 | 2 | Abdominal, foetal | Hospital | Poor |
| Pant (1995) | Case report | Nepal | Unknown | 1 | 1 | Foetal, other | Hospital | Poor |
| Rajagopalan (2015) | Case report | USA | Unknown | 1 | 1 | Foetal, other | Department of Anaesthesiology | Poor |
| Takeda (2013) | Case report | Japan | Unknown | 1 | 1 | Abdominal, other | Department of PS | Poor |
| Vathulya (2014) | Case report | India | Unknown | 1 | 1 | Foetal, other | Department of PS | Poor |
| Webb (1995) | Case report | Mexico | Unknown | 1 | 1 | Abdominal, other | Regional Burn Centre | Poor |

* = Midwives' notification system; ˆ = Plastic Surgery; ¥ Other includes: delivery mode, breastfeeding, and scar improvement.

The risk of bias assessment resulted in 21 out of 22 studies being classified as poor based on the Chambers criteria for case series; the cohort study received seven stars on three domains for nonrandomised studies, which means that the study was of good quality.

### 3.3. Patient and Burn Characteristics

Table 2 describes the characteristics of the patients per study. The 217 patients had a total of 330 pregnancies. The age at burn injury ranged from 3 to 27 years old. Some studies reported only 'childhood' for age at burn. The time window between burn injury and, most often, first pregnancy varied between 7 and 32 years. Most patients had a full-term pregnancy. In the studies that reported the Total Body Surface Area burned (TBSA), the range varied from less than 10 percent to more than 90 percent.

**Table 2.** Overview of patient characteristics.

| Study | Age at Burn, Yr | Age at First Pregnancy, Yr | Burn Characteristics | Duration of Pregnancy * |
|---|---|---|---|---|
| Duke (2012) | Mean 5.7 | Mean 20.9 | 131 patients <10% BSA 3 patients = 10–19% BSA | All: full term |
| Rai (1975) | Childhood | >15 yr or <45 yr | Full thickness: >4% of the abdomen | 35: full term; 2: premature labour; 3: abortion; (2: during pregnancy) |
| Kitzmiller (1998) | Mean 7.6 | ^ | Mean 55% BSA Full thickness: mean 42% | All: full term |
| Mitsukawa (2015) | ^ | ^ | Mean 64% BSA of the abdomen | ^ |
| McCauley (1990) | Mean 7.66 | Mean 19.83 | Mean 63.21% BSA Full thickness: mean 44.2% | All: full term |
| Daw (1983) | Mean 6.25 | Mean 20.5 | Around abdomen | 3: full term; 6: induction labour around full term; 2: premature labour |
| Matthews (1982) | ^ | ^ | Circumferential lower abdomen | ^ |
| Widgerow (1991) | Case 1: 9 Case 2: childhood | Case 1: 19 Case 2: 21 | Circumferential abdomen | Both: full term |
| Arabi (2019) | 5 | 20 | Chest, abdomen, upper limb, and part of her trunk | Full term |
| Aykan (2012) | 4 | 29 | Full thickness of genital region, bilateral lumber areas, lower two thirds of the abdominal wall | Full term |

**Table 2.** *Cont.*

| Study | Age at Burn, Yr | Age at First Pregnancy, Yr | Burn Characteristics | Duration of Pregnancy * |
|---|---|---|---|---|
| Cox (2015) | 2 | 31 | 2nd and 3rd degree burns on breasts, abdomen, thighs, lower back | Full term |
| Del Frari (2004) | 14 | 24 | Right lower abdomen, groin area, and thigh | ˆ (after 8 months) |
| Digregorio (1993) | 27 | 34 | 50% BSA third degree burns on face, hands, chest, and abdomen | Full term |
| Fioretti (1987) | 4 | 24 | From lower abdomen to thigh | Full term |
| Haeseker (1981) | 4 | 21 | Full thickness: 60% | Premature labour 3 days after operation |
| Kakagia (2012) | 3 | 30 | Postburn scars torso (anterior + lateral abdominal + chest wall, gluteal areas, breasts) | ˆ |
| Ozog (1963) | Childhood | 19 | From chest to midthigh | 1st pregnancy: 5 months; 2nd pregnancy: premature |
| Pant (1995) | 5 | 17 | Most of the perineum | ˆ |
| Rajagopalan (2015) | 6 | 38 | >90% BSA on chest, neck, face, abdomen, elbows, knees | 27 weeks |
| Takeda (2013) | 6 | 23 | BSA: 80% Full thickness: 65% | 36 weeks |
| Vathulya (2014) | 6 | 22 | Chest region to abdomen + perineal region with supra-clitoral hooding deformity; left breast nipple-areolar complex | Third trimester |
| Webb (1995) | 3 | 23 | Full thickness: mean 40% | Full term |

* = all pregnancies; ˆ = - this information is not provided in the publication.

Some of the studies wrote about the burns being specifically full thickness burns, second or third degree burns; however, most of the studies did not record the depth of the burns. The burn wound characteristics were mostly unclear: "around abdomen"; "most of the perineum"; "from chest to mid-thigh".

*3.4. Abdominal Burn Scar Complications*

Abdominal burn scar complications were reported in 14/22 studies (Table 3). The registry-based cohort study did not identify any admissions during pregnancies that were related to any (abdominal, breast, chest wall, back) scar complications or revisions of burn scars or contractures.

**Table 3.** Abdominal outcomes.

| Study | Complication | Follow Up | Outcome | Notes |
|---|---|---|---|---|
| Duke (2012) | No admissions during pregnancy for scar complications, revisions of scars or contractures; 2 times hypertrophic/ keloid scar was recorded | - | No long-term detrimental effects of burns on pregnancy, delivery or to the foetus | The majority of trunk burns were burns of partial thickness or unspecified depth |
| Rai (1975) | 2 itch, 7 tightness, 6 both | Unknown | Unknown | Three patients said the scars improved after pregnancy and in subsequent pregnancies |
| Kitzmiller (1998) | Minor scar breakdown in third trimester 25% instance of subjective sensation of abdominal tightness | Local care No narcotics necessary | Healed rapidly after delivery | |
| McCauley (1990) | Breakdown of abdominal scar tissue in 3$^{rd}$ trimester | Unknown | Unknown | |
| Daw (1983) | Tautness with a hot burning sensation to constant indescribable pain | Admitted to hospital, bed rest, inactivity, analgesics, surgical decompression (36 weeks) | Induction of labour in 6 of 11 pregnancies, premature labour | |
| Matthews (1982) | Maternal pain | Surgical intervention during 3rd trimester | Immediate pain relief | |
| Widgerow (1991) | Tightness, contracture limited progress of pregnancy | Surgical release (16 weeks/4 months) | Normal expansion of the uterus | |
| Aykan (2012) | Scar related hot burning sensation in 3rd trimester | Unknown | Unknown | Shortly after the operation, abdominal scar tissue tension-related symptoms and hot burning sensation decreased. |

**Table 3.** *Cont.*

| Study | Complication | Follow Up | Outcome | Notes |
|---|---|---|---|---|
| Cox (2015) | Intermittent ich and mild restriction with inactivity | Ablative functional laser (30 and 38 weeks) | Immediate postprocedure relief of tension, increased mobility, and improved respiration | Comfort and functionality were improved compared with prepregnancy; scar contour and pliability had improved |
| Fioretti (1987) | Mild dyspnoea | Had to reduce housework | Unknown | |
| Haeseker (1981) | Tightness and pain; potential obstruction for growing uterus | Surgical scar release (24 weeks) | Decompression→ premature labour→ foetus died | |
| Ozog (1963) | backache, nausea, anorexia, vomiting, dyspepsia, and severe constipation due to direct pressure | Re-examined each month and drug treatment | Refractory to drug treatment | |
| Takeda (2013) | Abdominal pain | Expansion abdominoplasty (20 weeks) | Abdominal wall expansion and foetal growth were found to be favourable | |
| Webb (1995 | Pain and a localised area of skin breakdown | Close monitoring | 38 weeks: pre-eclampsia→ CS | Taking advantage of the natural skin expansion of pregnancy |

The most reported abdominal burn scar complications reported in the other 13 studies were a feeling of tightness, pain, and itch. None of these studies reported on how these complications were assessed, or which measurement instrument or scale was used.

Five studies covering 49 patients had assessed tightness of the scar [6,16–19]. Approximately half of these patients ($n = 26/49$) indicated having complaints due to this tightness. Treatment for the tightness varied. In almost 84% ($n = 22/26$) of the cases, intervention was not deemed necessary ($n = 5/26$) or was not reported (17/26). Four of the 26 patients were unable to cope at home and were admitted for long-term bed rest.

Five studies covering eleven patients reported women with pain during pregnancy ($n = 10$), without describing details regarding where the pain was located [17,19–22]. Eight ($n = 8/11$) patients required surgical release due to abdominal pain. The scar release provided immediate pain relief. In one case [19], surgical release caused a very large gap, with a dramatic decompression effect as a result, and premature labour started at 24 weeks of gestation. One day after delivery, the newborn died due to prematurity. In two other studies, one case series and one case report, it was reported that two patients ($n = 2/13$) had pre-existing scar pain before pregnancy [23,24].

Itch of the scars during pregnancy was another complaint. In two studies, 9/22 (41%) patients reported experiencing (intermittent) itch of the scars [16,25]. One patient suffered from a restriction of daily activities because of the itch; therefore, she received ablative fractional laser resurfacing twice during pregnancy, which gave her immediate relief of abdominal tension and itch, increased mobility, and improved respiration.

Other abdominal burn scar complications included minor scar breakdown, reported in 3/27 patients from three studies [6,21,26], for which no intervention was needed. One case report [27] described a patient with mild dyspnoea due to the fact that the lower abdomen distension did not occur and the enlarged uterus was displaced to the upper abdomen; she had to reduce housework.

Other Symptoms

A number of single case studies reported symptoms such as a scar-related hot burning sensation [28], backache, nausea, constipation, and dyspepsia [29], where no intervention was reported and the association with burn scars was unlikely.

### 3.5. Foetal Complications

Five case studies described foetal complications, including pressure deformity and foetal death, in women with abdominal burn scars (Table 4). Foetal complications occurred during pregnancy [30–32] or after delivery [19,29]. In three of these case reports [30–32], it was described that the foetuses showed distress, although in one case (Rajagopalan et al.) it was unclear whether the foetal distress was caused by the abdominal burn scars of the mother. Two of the five foetuses died (Pant, Haeseker); in the case of Pant et al., the mother was already in labour for 22 h upon arrival in the hospital and could not deliver vaginally due to scar tissue, and the foetus had deceased; in the case described by Haeseker and Green, the newborn died due to premature labour after surgical release [19]. Ozog et al. reported on a case in which the foetus had a pressure deformity of the skull which gave the right side of the face a flat appearance and bilateral clubfoot deformity, which was due to little room for rotation and movement during pregnancy [29].

**Table 4.** Foetal outcomes.

| Study | Complication | Cause | Direct/Indirect Cause on Complication | Outcome | Likelihood of Relation to Burns |
|---|---|---|---|---|---|
| Haeseker (1981) | Premature labour | Surgical release→ decompression effect | Indirect | Dead due to prematurity | Likely/certain |
| Ozog (1963) | Pressure deformity of the skull→ right side of the face flat appearance + bilateral clubfoot deformity | Little room for rotation and movement | Direct | Temporary deformities | Likely |
| Pant (1995) | Non progressive labour (22hours in labour) | Scar tissue on perineum→ oedematous vulva with the foetal scalp visible | Direct | Dead | Likely/certain |
| Rajagopalan (2015) | Repeated foetal decelerations and non-reactive tracings | Unknown (preeclampsia, elevated aminotransferase, hyperglycaemia?) (placental insufficiency?) | Unknown | Emergency caesarean | Possible |
| Vathulya (2014) | Absent foetal heart sounds, meconium stained liquor, nonprogressive labour | Unknown | Unknown | Emergency caesarean | Possible |

In some of the other studies, foetal complications such as abortions, preterm labour, and a cleft palate were reported; however, the authors did not link this to the patients' abdominal burn scars.

*3.6. Other Complications*

3.6.1. Delivery Mode

A total of 15 studies reported (*n* = 208) the mode of delivery (Table 5). The registry-based cohort study did not observe a statistically significant difference in mode of delivery between subjects who had sustained a burn to the trunk (partial thickness, full thickness, or unspecified burn depth) and those who had sustained burns to other sites of the body or erythema burns to the trunk [33]. Together with Rai and MacG. Jackson, they conclude that the scarred abdominal wall does not seem responsible for different delivery modes [16,33]. There were several cases described in which the delivery mode was affected by burn scars. Five studies covering twenty-three patients [6,22,28,30,31] reported failure of labour due to cephalopelvic disproportion (*n* = 3/23) or perineal scar tissue (*n* = 4/23).

**Table 5.** Mode of Delivery.

| Study | Mode of Delivery | Potential Explanation | Notes |
|---|---|---|---|
| Duke (2012) | 142: NVD *; 26: instrument; 45: CS ^ | Unknown | No statistically significant differences between subjects who had sustained a burn to the trunk and those with burns to other sites of the body or erythema burns to trunk |
| Rai (1975) | 31: NVD; 4: forceps; 2: CS | Scarred abdominal wall was not responsible/Any lack of expulsive force not total excluded: an objective study by measurements of intraabdominal pressure changes and abdominal wall extensibility in relation to cervical dilatation is made | One of the forceps deliveries contained twins. Three abortions |
| Kitzmiller (1998) | 28: NVD; 3: CS | Failure of labour due to cephalopelvic disproportion | Abdominal wall healing after CS was not complicated |
| Mitsukawa (2015) | 2: NVD; 9: CS; 1: Not pregnant yet | If patients have scars covering 75% or more of the total abdominal area, scar release surgery is always performed. In addition, an open leg position is necessary. | |
| McCauley (1990) | 13: NVD; 1: CS | Unknown | 1 elective caesarean section |
| Daw (1983) | All: NVD | Abdominal pain from tightness | In 6 of 11 pregnancies necessitated |
| Aykan (2012) | CS | perineal scar tissue was dense and preventing vaginal delivery | classical Pfannenstiel incision was preferred |
| Cox (2015) | CS | Non progressive labour | 6 months after delivery she reported negligible tension and itch in the scarred areas |
| Fioretti (1987) | CS | the uterus could only expand transversely, foetal lie was transverse at term | Elective caesarean section |

**Table 5.** *Cont.*

| Study | Mode of Delivery | Potential Explanation | Notes |
|---|---|---|---|
| Ozog (1963) | 1st: stillborn, twins; 2nd: forceps | Unknown | 1 month premature |
| Pant (1995) | Incision anterior to the anus up to symphysis in the midline | Scar tissue covered most of the perineum | An incision was made anterior to the anus up to the symphysis pubis in the midline to separate the vulva obstruction. |
| Rajagopalan (2015) | CS | Foetal distress | Emergency caesarean section |
| Takeda (2013) | CS | Perineal scar contractures resulted in rigidity of the soft birth canal and limited hip joint flexion | Elective caesarean section |
| Vathulya (2014) | CS | Foetal distress and supra-clitoral hooding deformity: the clitoris, and the labia anterior 2/3 were almost invisible | Emergency caesarean section |
| Webb (1995) | CS | Preeclampsia and transverse lie | Caesarean section |

\* =Normal Vaginal Delivery; ˆ = Caesarean section.

In two case reports (*n* = 2), foetal lie was transverse at term [21,27]. In the case of Fioretti, the uterus only could expand transversely. Mitsukawa [23] noted that for the delivery method, sufficient extensibility is required in the infra-umbilical skin. In addition, an open leg position is necessary. If the scars cover 50% of more of this skin, caesarean section was desirable. Foetal distress and abdominal pain were two other reported reasons for a caesarean section or induction of labour. In four studies, it was not reported why a caesarean section was required.

3.6.2. Effects on Breastfeeding

Three studies included, covering eight patients, reported that five patients were not able to breastfeed due to damaged breast tissue [17,31,34]. In some cases, the patient was able to lactate from one breast. Two patients had enough lactation to feed their child.

*3.7. Positive Effects*

3.7.1. Scar Improvement

Hormonal changes during pregnancy have been thought to influence scars. Daw (1983) described that none of the six patients included in their case series reported any improvement in their scars [17]. Rai and MacG. Jackson found that 3/22 patients, either seen and examined or reviewed by questionnaire of unknown origin, experienced that the tightness and itching of the scars improved and that the scar became supple after their pregnancies. The rest noticed no change, and none of them said that the scars became worse [16].

3.7.2. Pregnancy as a Natural Tissue Expander

Four case reports described the possibility of using pregnancy after abdominal burns as a natural tissue expander method [21,24,35,36]. In three cases, reconstructive surgery was performed right after delivery to reconstruct an abdomen restricted by old burn scar tissue. In the fourth case, successful scar revision using artificial dermis and split-thickness skin grafting was performed in two stages, nine months before pregnancy. During

pregnancy, the grafted skin was extensively and naturally expanded by the gradually growing uterus [24].

## 4. Discussion

This review is the first to investigate potential complications during pregnancy after abdominal burn scars. In this review, we included 22 studies on pregnant women with abdominal scars due to burn injuries in the past. Pregnancies in these women may not be different from pregnancies in women without abdominal burn scars in regard to delivery mode and admissions for scar complications or revisions of burn scars or contractures. However, a certain proportion of these women may experience complaints such as tightness, pain, or itch. In these cases, few severe complications, including the need for surgery, were described. Positive effects have been described as well.

The observation that pregnancies in women with abdominal burn scars do not result in different outcomes compared with those in women who had sustained burns to other sites of the body or erythema burns to the trunk is based on one registry-based cohort study from Australia [33]. This study was based on clinically reported obstetric outcomes covered in the Midwives' Notification System rather than patient reported complications, such as pain, itch, and tightness. None of the other studies had a comparative nature. An important and interesting finding was that in some women the pregnancy had positive effects, including the experience of more supple scars. Oestrogen makes collagen looser [37]. Because a scar largely consists of collagen, increased levels of oestrogen during pregnancy may have positive effects on the scar, potentially making the scar more supple. In contrast, as mentioned before, in earlier literature, some cases reported worsening and recurrence of keloids and hypertrophic scars during pregnancy [8,9]. There is little evidence in the literature to support this phenomenon. A clinically relevant finding was the use of the expanding belly as a tissue expander. Pregnancy has been used as a tissue expander in the repair of a massive ventral hernia [38], where they used the gravid uterus as an intra-abdominal tissue expander. Although we did not perform a review on tissue expander techniques in women and our results are possibly not complete on this matter, we like to highlight the possibility of using pregnancy as a natural tissue expander method in women with abdominal burn scars; however, of course, ethical considerations regarding the patients' age should be kept in mind. Sustaining extensive burn injuries at a pre-pubertal age may stunt growth and influence breast formation, and often multiple scar releases are required.

A strength of our study was that it was prioritised questions and concerns raised by women with abdominal burn scars seen in our clinic and those who contacted the National burns information line from the Dutch Burns Foundation, which was one of the reasons we conducted this review. Another strength is the systematic method, which conforms to established guidelines on the conduct and reporting of reviews. Moreover, our literature search was performed in all major databases and was updated (until 23 April 2021) during the study to ensure the inclusion of the most recent studies. The current study also has some limitations. We only included studies written in English. As a result, we may have failed to capture specific scar aspects that are deemed important in non-English speaking countries. We did not use specific search terms (such as postnatal; breasts) which makes it possible that we did not include all possible complications in pregnant women after burn scars; however, this was not the primary aim of our study. The differences in design and outcome measures, such as the site of the burn injury and the severity of the burn, and the poor quality of the studies rendered the possibility of a quantitative analysis of the results, such as the incidence of various types of complications and dependency on burn severity, difficult. Finally, based on the design of the studies, publication bias is likely. The limited amount of evidence, the low quality of the studies, and the heterogeneity in outcomes mean that the findings of this review should be interpreted with some caution.

## 5. Conclusions

In conclusion, in this review, we observed indications that women with abdominal burn scars have normal obstetric outcomes; however, a certain proportion may experience mild to severe complications. Although limited, these data may be used as a first step to better inform women with abdominal burn scars, and it may contribute to personalised obstetric management. Although this perhaps concerns a rare subgroup of patients, the impact of burns on quality of life can be tremendous, even after so many years. The results of this study might also create awareness of the futuristic child wish in the acute treatment of severe abdominal, thoracic, and genital burns in girls and young women. More quantitative and qualitative research is required to assess the accurate incidence, type, and predictive factors of complications during and related to pregnancy in women with abdominal burn scars. This should also include an assessment of foetal outcomes.

**Author Contributions:** Conceptualization, Z.J.v.G., A.S. and A.P.; methodology, Z.J.v.G., A.S., R.d.V. and A.P.; formal analysis, Z.J.v.G., A.S. and A.P.; data curation, Z.J.v.G., A.S. and A.P.; writing—original draft preparation, Z.J.v.G. and A.S.; writing—review and editing, Z.J.v.G., A.S., P.P.M.v.Z., R.d.V. and A.P.; supervision, P.P.M.v.Z. and A.P.; project administration, Z.J.v.G., A.S. and A.P.; funding acquisition, A.P. All authors have read and agreed to the published version of the manuscript.

**Funding:** This research received no external funding. Funding for open access publication was provided by the Dutch Burns Foundation.

**Conflicts of Interest:** The authors declare no conflict of interest.

## Appendix A

**Table A1.** PubMed Session Results (23 April 2021).

| Search | Query | Items Found |
|---|---|---|
| #8 | #4 OR #7 | 1388 |
| #7 | #5 AND #6 | 248 |
| #6 | "pregnan*"[ti] OR "pregnan*"[ot] OR "vaginal"[ti] OR "vaginal"[ot] OR "abdominal"[ti] OR "abdominal"[ot] OR "truncal"[ti] OR "truncal"[ot] | 402,641 |
| #5 | "burn"[ti] OR "burns"[ti] OR "scald*"[ti] OR "postburn *"[ti] OR ("thermal"[ti] AND "injur*"[ti]) OR ("chemical"[ti] AND "injur *"[ti]) OR "burn"[ot] OR "burns"[ot] OR "scald*"[ot] OR "postburn *"[ot] OR ("thermal"[ot] AND "injur*"[ot]) OR ("chemical"[ot] AND "injur*"[ot]) | 44,069 |
| #4 | #1 AND #2 AND #3 | 1153 |
| #3 | "abdom*"[tw] OR "truncal"[tiab] | 428,002 |

**Table A1.** *Cont.*

| Search | Query | Items Found |
|---|---|---|
| #2 | "Pregnancy Complications"[Mesh] OR "Gestational Age"[Mesh] OR "Pregnancy"[Mesh] OR "Pregnancy Trimesters"[Mesh] OR "Pregnant Women"[Mesh] OR "Preconception Care"[Mesh] OR "Maternal Mortality"[Mesh] OR "Maternal Health"[Mesh] OR "Fetal Mortality"[Mesh] OR "Delivery, Obstetric"[Mesh] OR "maternal"[tiab] OR "mother*"[tiab] OR "fetal"[tiab] OR "foetal"[tiab] OR "fetus"[tiab] OR "foetus"[tiab] OR "maternity"[tiab] OR "pregnan*"[tiab] OR "pseudopregnan*"[tiab] OR "gravidit*"[tiab] OR "nulligravid*"[tiab] OR "primigravid*"[tiab] OR "multigravid*"[tiab] OR "gravidation"[tiab] OR "gravidarum"[tiab] OR "gravida"[tiab] OR "parturition*"[tiab] OR "parity"[tiab] OR "childbirth*"[tiab] OR "birthing"[tiab] OR "birth"[tiab] OR "stillbirth"[tiab] OR "childbed"[tiab] OR ("abdominal"[tiab] AND "deliver*"[tiab]) OR "gestation*"[tiab] OR "parturien*"[tiab] OR "child-bear*"[tiab] OR "childbear*"[tiab] OR "placentat*"[tiab] OR "prepregnan*"[tiab] OR "conception*"[tiab] OR "preconception*"[tiab] OR "obstetric*"[tiab] OR "prenatal"[tiab] OR "perinatal"[tiab] OR "intranatal"[tiab] OR "antenatal"[tiab] OR "prepartum"[tiab] OR "peripartum"[tiab] OR "intrapartum"[tiab] OR "antepartum"[tiab] OR "pre-natal"[tiab] OR "peri-natal"[tiab] OR "intra-natal"[tiab] OR "ante-natal"[tiab] OR "pre-partum"[tiab] OR "peri-partum"[tiab] OR "intra-partum"[tiab] OR "ante-partum"[tiab] | 1,625,975 |
| #1 | "Cicatrix"[Mesh] OR "cicatr*"[tiab] OR "keloid*"[tiab] OR "scar"[tiab] OR "scars"[tiab] OR "scarring"[tiab] OR "contractur*"[tiab] | 123,866 |

**Table A2.** Embase.com Session Results (23 April 2021).

| Search | Query | Items Found |
|---|---|---|
| #9 | #8 NOT ('conference abstract'/it OR 'conference review'/it) | 1778 |
| #8 | #4 OR #7 | 2433 |
| #7 | #5 AND #6 | 274 |
| #6 | 'pregnan*':ti OR 'pregnan*':kw OR 'vaginal':ti OR 'vaginal':kw OR 'abdominal':ti OR 'abdominal':kw OR 'truncal':ti OR 'truncal':kw | 511,336 |
| #5 | 'burn':ti OR 'burns':ti OR 'scald*':ti OR 'postburn*':ti OR ('thermal':ti AND 'injur*':ti) OR ('chemical':ti AND 'injur*':ti) OR 'burn':kw OR 'burns':kw OR 'scald*':kw OR 'postburn*':kw OR ('thermal':kw AND 'injur*':kw) OR ('chemical':kw AND 'injur*':kw) | 53,170 |
| #4 | #1 AND #2 AND #3 | 2171 |
| #3 | 'abdom*':ab,ti,kw,de OR 'truncal':ab,ti,kw | 769,423 |
| #2 | 'pregnancy complication'/exp OR 'gestational age'/exp OR 'pregnancy'/exp OR 'named groups by pregnancy'/exp OR 'prepregnancy care'/exp OR 'maternal mortality'/exp OR 'maternal welfare'/exp OR 'fetus mortality'/exp OR 'fetal health'/exp OR 'obstetric delivery'/exp OR 'maternal':ab,ti,kw OR 'mother*':ab,ti,kw OR 'fetal':ab,ti,kw OR 'foetal':ab,ti,kw OR 'fetus':ab,ti,kw OR 'foetus':ab,ti,kw OR 'maternity':ab,ti,kw OR 'pregnan*':ab,ti,kw OR 'pseudopregnan*':ab,ti,kw OR 'gravidit*':ab,ti,kw OR 'nulligravid*':ab,ti,kw OR 'primigravid*':ab,ti,kw OR 'multigravid*':ab,ti,kw OR 'gravidation':ab,ti,kw OR 'gravidarum':ab,ti,kw OR 'gravida':ab,ti,kw OR 'parturition*':ab,ti,kw OR 'parity':ab,ti,kw OR 'childbirth*':ab,ti,kw OR 'birthing':ab,ti,kw OR 'birth':ab,ti,kw OR 'stillbirth':ab,ti,kw OR 'childbed':ab,ti,kw OR ('abdominal':ab,ti,kw AND 'deliver*':ab,ti,kw) OR 'gestation*':ab,ti,kw OR 'parturien*':ab,ti,kw OR 'child-bear*':ab,ti,kw OR 'childbear*':ab,ti,kw OR 'placentat*':ab,ti,kw OR 'prepregnan*':ab,ti,kw OR 'conception*':ab,ti,kw OR 'preconception*':ab,ti,kw OR 'obstetric*':ab,ti,kw OR 'prenatal':ab,ti,kw OR 'perinatal':ab,ti,kw OR 'intranatal':ab,ti,kw OR 'antenatal':ab,ti,kw OR 'prepartum':ab,ti,kw OR 'peripartum':ab,ti,kw OR 'intrapartum':ab,ti,kw OR 'antepartum':ab,ti,kw OR 'pre-natal':ab,ti,kw OR 'peri-natal':ab,ti,kw OR 'intra-natal':ab,ti,kw OR 'ante-natal':ab,ti,kw OR 'pre-partum':ab,ti,kw OR 'peri-partum':ab,ti,kw OR 'intra-partum':ab,ti,kw OR 'ante-partum':ab,ti,kw | 1,958,693 |
| #1 | 'scar'/exp OR 'cicatr*':ab,ti,kw OR 'keloid*':ab,ti,kw OR 'scar':ab,ti,kw OR 'scars':ab,ti,kw OR 'scarring':ab,ti,kw OR 'contractur*':ab,ti,kw | 167,645 |

**Table A3.** Scopus Session Results (23 April 2021).

| Search | Query | Items Found |
|---|---|---|
| #8 | #4 OR #7 | 1132 |
| #7 | #5 AND #6 | 278 |
| #6 | TITLE ("pregnan*" OR "vaginal" OR "abdominal" OR "truncal") OR AUTHKEY ("pregnan*" OR "vaginal" OR "abdominal" OR "truncal") | 498,977 |
| #5 | TITLE ("burn" OR "burns" OR "scald*" OR "postburn*" OR ("thermal" AND "injur*") OR ("chemical" AND "injur*")) OR AUTHKEY ("burn" OR "burns" OR "scald*" OR "postburn*" OR ("thermal" AND "injur*") OR ("chemical" AND "injur*")) | 64,253 |
| #4 | #1 AND #2 AND #3 | 865 |
| #3 | TITLE-ABS ("abdom*" OR "truncal") OR AUTHKEY ("abdom*" OR "truncal") | 463,391 |
| #2 | TITLE-ABS ("maternal" OR "mother*" OR "fetal" OR "foetal" OR "fetus" OR "foetus" OR "maternity" OR "pregnan*" OR "pseudopregnan*" OR "gravidit*" OR "nulligravid*" OR "primigravid*" OR "multigravid*" OR "gravidation" OR "gravidarum" OR "gravida" OR "parturition*" OR "parity" OR "childbirth*" OR "birthing" OR "birth" OR "stillbirth" OR "childbed" OR ("abdominal" AND "deliver*") OR "gestation*" OR "parturien*" OR "child-bear*" OR "childbear*" OR "placentat*" OR "prepregnan*" OR "conception*" OR "preconception*" OR "obstetric*" OR "prenatal" OR "perinatal" OR "intranatal" OR "antenatal" OR "prepartum" OR "peripartum" OR "intrapartum" OR "antepartum" OR "pre-natal" OR "peri-natal" OR "intra-natal" OR "ante-natal" OR "pre-partum" OR "peri-partum" OR "intra-partum" OR "ante-partum") OR AUTHKEY ("maternal" OR "mother*" OR "fetal" OR "foetal" OR "fetus" OR "foetus" OR "maternity" OR "pregnan*" OR "pseudopregnan*" OR "gravidit*" OR "nulligravid*" OR "primigravid*" OR "multigravid*" OR "gravidation" OR "gravidarum" OR "gravida" OR "parturition*" OR "parity" OR "childbirth*" OR "birthing" OR "birth" OR "stillbirth" OR "childbed" OR ("abdominal" AND "deliver*") OR "gestation*" OR "parturien*" OR "child-bear*" OR "childbear*" OR "placentat*" OR "prepregnan*" OR "conception*" OR "preconception*" OR "obstetric*" OR "prenatal" OR "perinatal" OR "intranatal" OR "antenatal" OR "prepartum" OR "peripartum" OR "intrapartum" OR "antepartum" OR "pre-natal" OR "peri-natal" OR "intra-natal" OR "ante-natal" OR "pre-partum" OR "peri-partum" OR "intra-partum" OR "ante-partum") | 1,957,453 |
| #1 | TITLE-ABS ("cicatr*" OR "keloid*" OR "scar" OR "scars" OR "scarring" OR "contractur*") OR AUTHKEY ("cicatr*" OR "keloid*" OR "scar" OR "scars" OR "scarring" OR "contractur*") | 141,659 |

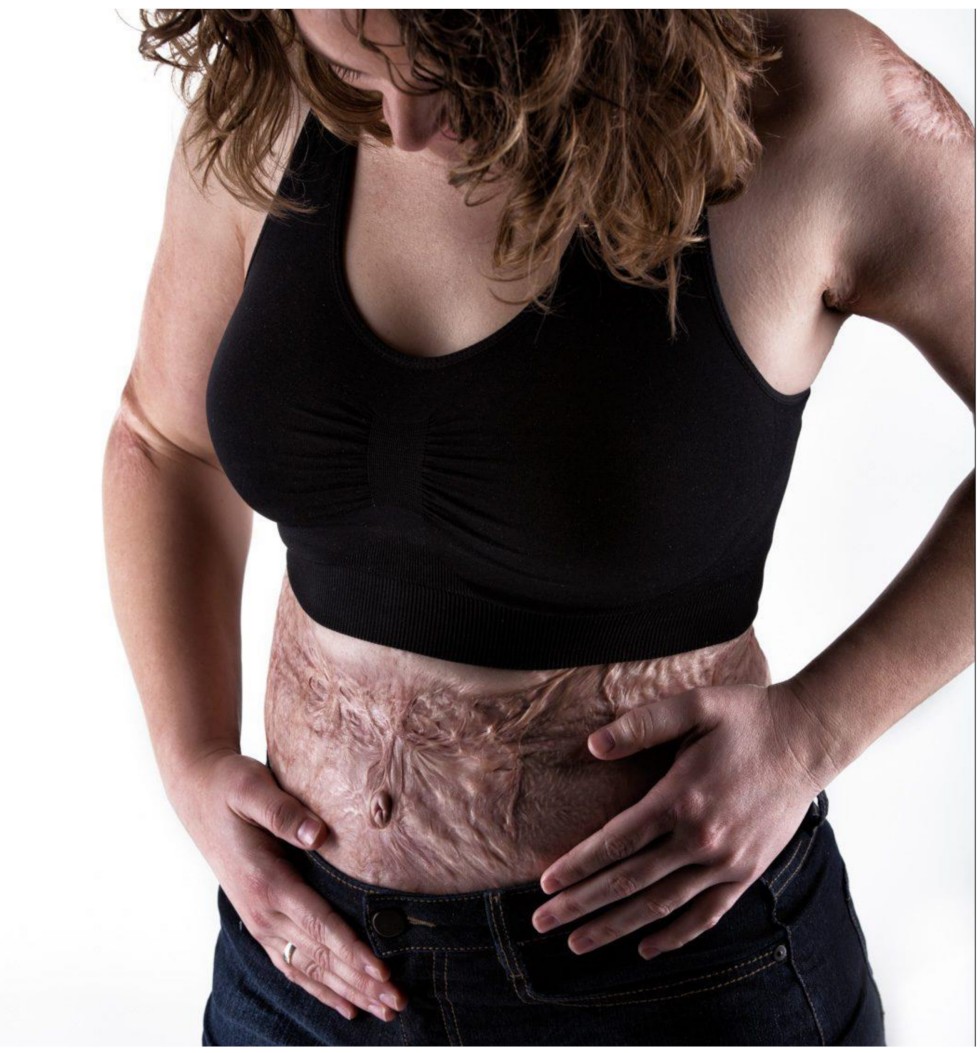

**Figure A1.** Photo by: Jan van Beijnhem, Foto Studio XL. Written permission from patient and photographer.

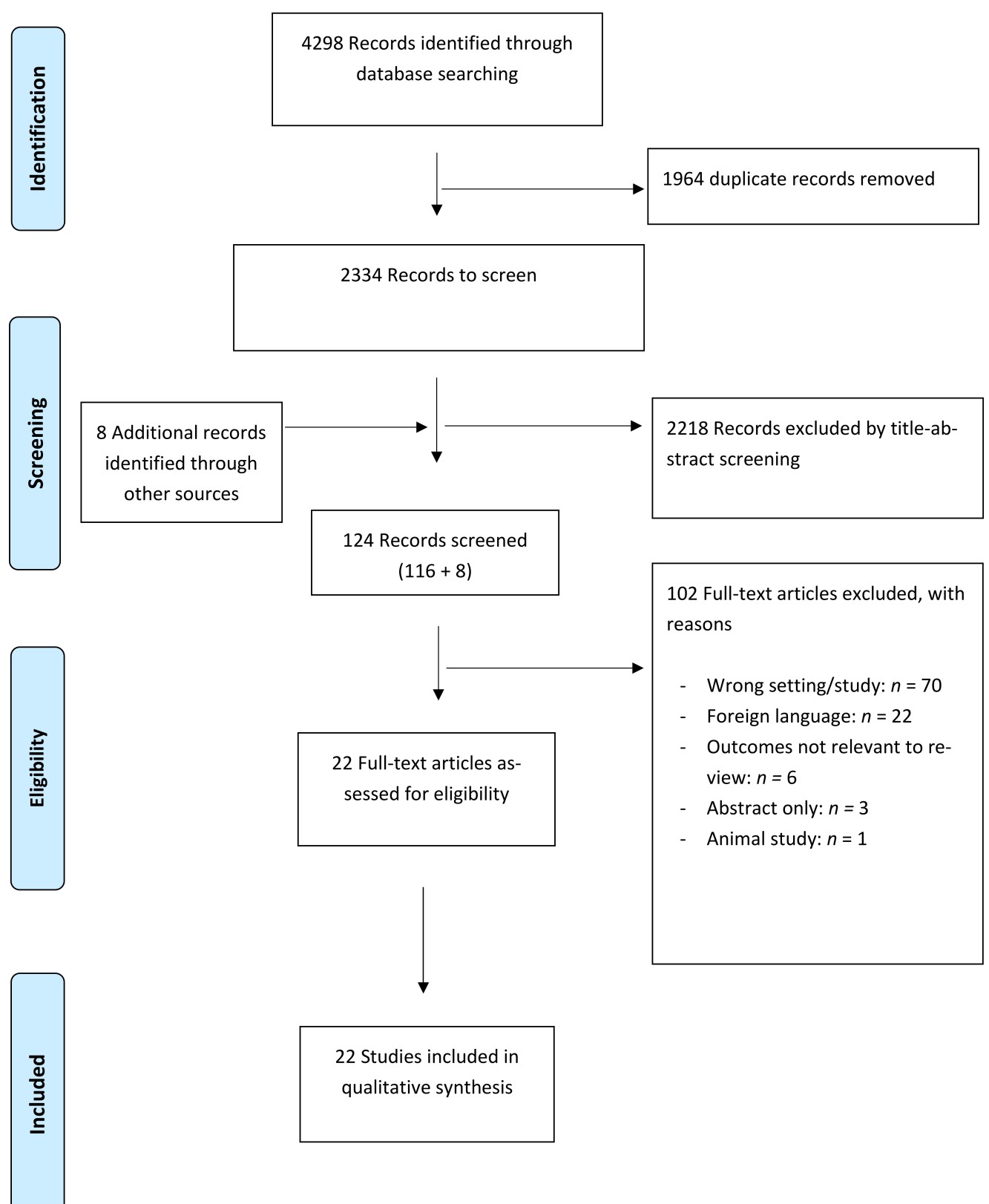

**Figure A2.** PRISMA flow diagram.

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
