# Peer review of "Complications during Pregnancy after Abdominal Burn Scars: A Review"

_2673-1991, doi:10.3390/ebj4010005_

Round 1

Reviewer 1 Report

This is a well-written review that summarized the complications of the pregnant with abdominal scar and the potential impact of abdominal scar on obstetric outcome. It is worthwhile to share the summarized data with the readers. However, in addition to the complications and the delivery mode, the potential impact of abdominal scar, as the restricted space might limited the expansion of uterus, on the quality of fetus as well as neonate is important information. It will be much better if the authors could add, if there are, the information of evaluation of fetus and naonate quality.

Reviewer 2 Report

Thank for you the opportunity to review this manuscript “Complications during pregnancy after abdominal burn scars: A review”. This is a very interesting paper; however, the methodology section requires major review.  The results section requires revision including revision of formatting. However, there is a good balance of text and tables. The discussion requires major revision as the current manuscript does not support these findings. This conclusion requires minor revision to include appropriate recommendations for future research.

Abstract

Based on the feedback provided in Materials and Methods- this section requires review to reflect the changes.

Introduction

This introduction provides an overview and rationale for the review. Line 41 burn survival improved – should read as burn survival has improved.

This recent study should be discussed to further support the aim of this current study.

Dijkerman ML, Breederveld-Walters ML, Pijpe A, Breederveld RS. Management and outcome of burn injuries during pregnancy: a systematic review and presentation of a comprehensive guideline. Burns. 2022 Apr 4.

I would recommend revising the final line of the introduction to make it clear this is the aim.

Materials and Methods

This section requires major revision. Please provide more clarity around the time frames for the search strategy in this study, this is not transparent, nor does it match, to the information registered on PROSPERO

Anticipated or actual start date

22 May 2020

Anticipated completion date

06 July 2022

Date of registration in PROSPERO

18 June 2022

Date of first submission

07 June 2022

Thank you for providing the BOOLAN phrases in Appendix A, as above the correct date ranges for the searches is required to validate the results.  

The process for reviewing the studies is clear.

Line 95 injuries in the past; (iii) fetal complications (iiii) other complications due to burns. (iiii) is not correct (iv) is the correct numerical system.

The inclusion and exclusion criteria have been defined, however please provide more detail as to why burns obtained during pregnancy have been excluded. If this is due to the following study:

Dijkerman ML, Breederveld-Walters ML, Pijpe A, Breederveld RS. Management and outcome of burn injuries during pregnancy: a systematic review and presentation of a comprehensive guideline. Burns. 2022 Apr 4.

Please add this to the introduction to increase the trustworthiness of this manuscript.

The method for reducing bias has been clearly outlined, as was the method for study analysis/ synthesis.  

Results

Literature search: – this section requires major revision.

Figure A1 is not referenced at all in this manuscript.

Line 1-Figure 1 should come before Figure 2

The figure reference should match the transcript – Line 130 Figure 2 page 14 A2

Figure A2 page 14 has errors – 4298 articles were found 2334 were duplicates leading to 1964 potential articles – however the table states 2218 were excluded. The table also states 8 additional records were identified from other sources- this is not discussed in the body of the manuscript and should be to aid clarity for the reader.  

Study and patient characteristics:

Please add what the other is referring too under outcome measures Table A1

Abdominal burn scar complication:

Please clarify how scar tightness/pain was assessed and how this was differentiated from abdominal tightness/pain during pregnancy which can be caused by ligament stretching. Please demonstrate evidence for   scar scales if these were used.

Line 179 – typo please correct

Line 180 and 181- please clarify if this pain was due to the burn scar and how long post burn were the participants.

Paragraph commencing line 182 – please clarify if laser therapy was included to the abdomen.

Line 192 -194 should be under a separate heading as they cannot be attributed to the burn scar.  

Positive effects

Please indicate if scar assessments were used or how these scar changes were recorded. For example, only recorded by the treating team, patient reports etc.     

Discussion

The findings of this paper are mostly positive however the first line (253) of the discission is negative.  Once the aim has been revised as advised earlier, this first sentence should be re phrased to align.

Line 262 given that the information was sought from the MNS data base as referenced in Table A1 it is highly unlike that burn scar outcomes would be collected here. Please discuss further.

While pregnancy may act as a natural tissue expander for abdomen burn scars what are the ethical concerns regarding this for young people who sustain burns pre puberty? This requires further consider consideration, discussion, and clarity.   

Line 282 states this study is patient based. This is the first time this has been mentioned. Please see the feedback for the introduction and aims. This statement is not supported based on the current manuscript.

As the methodology section required major revision, these statements in the discussion cannot be supported.

The limitations are appropriate.

Conclusion

The conclusion is reflective of the small numbers who have complications in pregnancy post abdominal burns.  However, in order to truly understand the impact this has on individuals’ in-depth qualitative interviews are required. I would strongly suggest this recommendation be included to explore this topic further to improve outcomes long term which are patient centred. 

Reviewer 3 Report

The authors of Complications during pregnancy after abdominal burn scars: A review performed a review of the literature to identify abdominal, fetal and potential other complications during pregnancy in women with abdominal burn scars. They included and reviewed 22 studies comprising 217 patients finding that overall, complications during pregnancy after abdominal burn scars were limited. They found that the time between burn injury and first pregnancy varied between 7-32 years and that most of the women had normal pregnancies. They reported that the most common complaints were increased feeling of tightness, itch, and/of pain or scar breakdown. While there were some cases of fetal complications in the literature, these cases were limited. This was a well done study. There were some minor spelling errors, but the message remained clear.

On page 4, line 144-145 is unclear. Is “that” missing between studies and were.

This could be of real value to the burn survivor community.

Round 2

Reviewer 2 Report

Manuscript Title: Complications during pregnancy after abdominal burn scars: A review

Overall

Thank for you the opportunity to review this revised manuscript “Complications during pregnancy after abdominal burn scars: A review”.

While the authors addressed the reviewer’s feedback in their response letter, major changes have not been made within the manuscript. Unfortunately, further revisions are required to bring this to the standard of the journal.

Abstract

This has not been sufficiently addressed.

Introduction

Thank you for revising the aim, this is now clear for the reader.

Materials and Methods

This section continues to require major revision. While I understand you have contacted PROSPRO from your response, this is not clear for the reader, and this should be more transparent in the manuscript. Please revise.

Results

Literature search: – this section requires major revision.

Figure A2 page 14 continues to have errors – 4298 articles were found 2334 were duplicates leading to 1964 potential articles – however the table continues to state 2218 were excluded.

The numbers have been revised in this section with no explanation in the text of the manuscript. Again, the authors have addressed this in their response to the reviewer but not in the manuscript. The process followed to include/exclude articles remain unclear in this table.  This section undermines the manuscript due to lack of transparency.

Abdominal burn scar complication:

Thank you for clarifying that no scar scales were used.

Table A2 does not have any revisions on version 2 – unknown continues to be used. Please revise.

Line 198 -200 should be under a separate heading as they cannot be attributed to the burn scar.  The authors have addressed this in their response to the review and agreed with the reviewer yet have not changed this. This is misleading and should have a separate heading even if for one line.  

Discussion

While pregnancy may act as a natural tissue expander for abdomen burn scars what are the ethical concerns regarding this for young people who sustain burns pre puberty? If an individual has a burn pre – puberty it is well documented that this stunts growth including breast formation, they require multiple scar releases please discuss.  

Thank you, the statement, regarding patient based is NOT referring to animal studies rather referring to patient based, or patient led. Clarity around anecdotal feedback has clarified this.

The authors have not revised this section sufficiently.

Conclusion

This section has been revised sufficiently.  
